# Development and Characterization of Refractance Window-Dried *Curcuma longa* Powder Fortified with NaFeEDTA and Folic Acid: A Study on Thermal, Morphological, and In Vitro Bio Accessibility Properties

**DOI:** 10.3390/foods14040658

**Published:** 2025-02-15

**Authors:** Preetisagar Talukdar, Kamal Narayan Baruah, Pankaj Jyoti Barman, Sonu Sharma, Ramagopal V. S. Uppaluri

**Affiliations:** 1Department of Chemical Engineering, Indian Institute of Technology Guwahati, Guwahati 781039, Assam, India; ptalukdar1@rgu.ac (P.T.); ramgopalu@iitg.ac.in (R.V.S.U.); 2Department of Food Technology, Assam Royal Global University, Guwahati 781035, Assam, India; 3Department of Biotechnology, Sharda School of Engineering and Technology, Sharda University, Greater Noida 201310, Uttar Pradesh, India; 4Department of Community Medicine, Gauhati Medical College, Guwahati 781032, Assam, India; pankajbarman6411@gmail.com; 5Research & Development Department, Cambridge Treats Inc., 115 Goddard Cres, Cambridge, ON N3E 0B1, Canada; sonucanada45@gmail.com

**Keywords:** turmeric, curcumin, fortification, folic acid, NaFeEDTA, bio-accessibility

## Abstract

*Curcuma longa* powder was prepared by refractance window drying (RWD) and was fortified. Fortification of dried turmeric powder with folic acid and NaFeEDTA, along with its characterization, was achieved. Characterization techniques, such as FTIR, XRD, TGA, DSC, FESEM, and particle size analysis, have been considered to study the morphological, thermal, and crystalline properties of the resulting fortified turmeric. In vitro digestion studies were carried out to determine the retention of nutrients after fortification. The RW-dried and fortified turmeric powder exhibited a stable average particle size and PDI values in the range of 1500–1600 nm, for 0.25–0.29, respectively. The fortified turmeric powder exhibited enhanced crystalline properties with sharp and high intensity peaks for NaFeEDTA-fortified turmeric powder. In vitro digestion studies affirmed the bio-accessibility of the novel fortified turmeric powder at 9.77 mg/100 g and 12.74 mg/100 g for folic acid and NaFeEDTA fortification cases, respectively. Thus, the findings confirmed that there was no significant influence of fortification on the characteristics of folic acid and the NaFeEDTA-fortified RW-dried turmeric powder product.

## 1. Introduction

Vitamin and mineral deficiencies lead to a variety of health problems, including learning disabilities, intellectual disability, reduced work capacity, blindness, and even premature death [1,2,3]. In order to address these issues, food fortification has emerged as the preferred solution with respect to traditional pharmaceutical supplements [1]. This process involves enriching foods with essential micronutrients like vitamins, minerals, and trace elements to combat multiple nutrient deficiencies and improve health outcomes without significant risks to health. Commonly fortified foods to help alleviate these deficiencies include cereal, flour, rice, and milk. However, fortification poses challenges such as nutrient bio-accessibility, undesirable organoleptic changes, and the potential rejection of the fortified product by consumers [4].

Three primary micronutrient deficiencies have been identified as major concerns: iron, iodine, and vitamin deficiencies [1]. Folate has gained significant global attention due to its critical role in preventing early embryonic brain development disorders, such as neural tube defects (NTDs) [5,6]. NTDs, arising from the incomplete closure of the neural tube within the first 4 weeks of conception, are a major cause of serious congenital disorders, affecting 0.2 to 10 out of 1000 pregnancies worldwide. While spina bifida, anencephaly, and encephalocele are the most common types of NTDs, iniencephaly and craniorachischisis are rare but also included [7]. Maternal folate deficiency during early pregnancy is a preventable risk factor. The World Health Organization (WHO) recommends that women of reproductive age maintain red blood cell folate concentrations above 400 ng/mL to minimize NTD risk [8]. Preventative measures include folic acid supplementation (0.4 mg daily before conception until the first trimester), consumption of folate-rich foods (leafy greens, asparagus, beets, broccoli, and artichokes), and fortified foods. For the prevention of NTDs, fortification of staple foods and ingredients such as wheat flour, maize flour, and rice with folic acid has been implemented in nearly 60 countries as a safe and cost-effective public health strategy [9].

Throughout life, iron requirements vary with developmental stages. Dietary iron alone may not always meet changes in iron demand. Consequently, this necessitates oral iron supplementation and food fortification. Adequate iron intake is crucial, as iron deficiency anemia in infancy has been linked to lasting neurodevelopmental and cognitive delays. This is valid even after restoration of iron levels [10]. The WHO advises administering supplementary iron to infants and toddlers aged 6–23 months, particularly in areas where anemia prevalence exceeds 40%, and for the prevention and treatment of iron deficiency and iron deficiency anemia. A daily dose of 10–12.5 mg of elemental iron for 3 months is the recommended approach [11]. A study investigated the potential of *Gryllus assimilis* (cricket) powder, alone and in combination with soy flour, for the improvement of iron bio-accessibility in rodents. While overall food intake, weight gain, and bio-accessibility measures were similar across groups, the combination of cricket powder and soy flour led to higher hemoglobin levels in comparison to cricket powder alone. These results suggest that a combination of *Gryllus assimilis* with soy flour may offer enhanced benefits for iron bio-accessibility and overall health [12].

In India, where 79% of children aged 6 to 35 months and women aged 15 to 49 years are anemic [13], iron deficiency is primarily attributed to the consumption of foods with low iron bio-accessibility. Several products, such as salt, wheat, corn, and rice, have been fortified with iron [13,14]. However, iron has been the most difficult micronutrient to incorporate into food products. This is due to differences in iron absorption rates. Because of this, undesirable variations in flavor and color of the fortified foods are inevitable. Another issue could be that the inhibitors present in various food matrices could retard iron absorption. Despite such challenges, technologies supporting the large-scale fortification of wheat and corn flour are well-established [14]. A recently developed extrusion-based premix technique for rice fortification has demonstrated significant potential [14]. However, many alternate iron-fortified food products are to be developed to affirm upon the availability of choices and varieties for a wide range of global population.

*Curcuma longa*, widely known as turmeric, has been extensively studied for its health-promoting properties, and especially for its key active component, i.e., curcumin [15]. Curcumin, a polyphenolic, has anti-inflammatory, antioxidant, and anticancer effects [16]. Curcumin reduces inflammation through the targeting of multiple pathways and substances. Notably, it inhibits the nuclear factor kappa B (NF-κB) pathway, a major contributor to inflammation [17]. Curcumin also downregulates the production of inflammatory chemical compounds, tumor necrosis factor α, and enzymes. Furthermore, it acts as an antioxidant and eliminates free radicals, reduces pain perception through antinociceptive actions, promotes tissue regeneration, and, additionally, aids in wound healing [18].

However, despite its therapeutic potential, the clinical application of curcumin is significantly limited. This is due to its poor bio-accessibility. Various strategies have been explored to enhance the absorption and efficacy of curcumin. These include the utilization of adjuvants such as piperine. Yet, these methods often yield inconsistent results and may not provide a comprehensive solution to the challenge of curcumin’s bio-accessibility [19].

Recent advancements in food fortification have opened new avenues for improved nutritional profiles in functional foods. The incorporation of essential micronutrients such as sodium iron ethylenediaminetetraacetic acid (NaFeEDTA) and folic acid into *Curcuma longa* powder presents a promising approach. NaFeEDTA is recognized for its high bio-accessibility as an iron source and has been recommended for the fortification of various food products to combat iron deficiency anemia [20]. Folic acid, on the other hand, plays a critical role in the prevention of neural tube defects and in overall health enhancement, especially during pregnancy [21]. The fortification of turmeric powder with these micronutrients is not only aimed at improving its nutritional value, but is also done in order to enhance the health benefits linked with curcumin consumption. Various studies have investigated the effects of fortificants on mineral-fortified dried products. For example, in a relevant prior study, authors fortified Nepalese curry powder with different iron compounds [22]. In another work, the authors enriched whole wheat flour with a fortificant premix consisting of ferrous sulfate, ethylenediamine tetra-acetic acid (EDTA), and folic acid [23]. Some studies have also addressed salt fortification using folic acid, iron, and iodine [13]. Other researchers have addressed the fortification of whole wheat flour with various iron compounds, including folic acid [24,25]. Additionally, researchers fortified chickpea seeds and flour with ferrous sulfate heptahydrate, ferrous sulfate monohydrate, and NaFeEDTA fortificants [26]. Similarly, reports conveying the utilization of finger millet and sorghum flours as double fortification vehicles with ferrous fumarate, zinc stearate, and EDTA have been presented [27].

Refractance Window (RW) drying is a gentle, advanced, and energy-efficient process, which employs a thin-film drying system coupled with water recirculation for quick and effective moisture removal from food materials. In this process, a food product is spread in a thin layer on a transparent plastic conveyor belt that floats on a heated water surface. The “refractance window” effect occurs as infrared energy is transmitted through the belt and thereby prompts the removal of water from the product, which constitutes the same. This leads to efficient drying at lower product temperatures (<80 °C). In comparison to conventional drying methods, the RWD technique offers significant advantages in terms of final product quality, including improved texture, color retention, and nutrient preservation, as well as lower energy consumption and reduced drying times [28,29,30]. However, RW drying has numerous challenges in comparison to the other drying methods. These include limited throughput, contingency on water availability, high initial costs, and surface-level drying, rather than deep penetration, in food. Despite these constraints, the RW drying process presents affordable potential for the large-scale production of high-quality food products.

So far, parametric optimization of the RW drying process has been primarily focused on carrots and onions, and little attention has been paid towards fortified versions of other foods [29,31]. In a few studies, the RW drying technique has been applied for the preservation of bioactive compounds in *Curcuma longa* [29,32]. This also includes our earlier work, wherein we optimized the RW process for *Curcuma longa* [32]. However, extensive studies on turmeric being prepared with RW drying in terms of product fortification are limited. Thus, we hypothesized that there is a possibility to develop new functional and bio- accessible fortified turmeric products. Keeping this in mind, our hypothesis will be tested with a comprehensive assessment of the RW-dried turmeric powder along with its folic acid- and NaFeEDTA-fortified variants in terms of thermal stability, morphological characteristics, and in vitro bio-accessibility. The properties of the developed product will be evaluated with specific analytical techniques such as thermogravimetric analysis (TGA), differential scanning calorimetry (DSC), field-emission scanning electron microscopy (FESEM), X-ray diffraction (XRD), Fourier transform infrared spectroscopy (FTIR), and particle size analysis. The developed turmeric product has prospects as a nutritional source for a large section of the worldwide population.

## 2. Materials and Methods

### 2.1. Raw Materials, Chemicals, and Sample Preparation

The local turmeric variety (*Curcuma longa*), cultivated in Assam, India, was sourced from a vendor at the market complex in Indian Institute of Technology Guwahati, Kamrup, Assam, India. Thereafter, it was packaged in a polyethylene pouch to prevent contamination during transportation. Sodium ferric ethylenediaminetetraacetate (NaFeEDTA), folic acid, potassium bromide (KBr), sodium chloride (NaCl), potassium chloride (KCl), hydrochloric acid (HCl), sodium bicarbonate (NaHCO_3_), calcium chloride (CaCl_2_), sodium hydroxide (NaOH), pepsin, pancreatin, bile salts, and curcumin were obtained from Sigma Aldrich India. The raw turmeric was washed with tap water to remove surface contaminants, wiped with tissue paper to eliminate excess water, peeled, and then sliced to a thickness of 1 mm with an adjustable slicer (Model: Ganesh Slicer, GM Industries, Vasai, India). Thereafter, the sliced samples were subjected to RWD.

### 2.2. Refractance Window Drying

The RWD process was carried out in batches, with optimized parameters from prior experiments, i.e., a sample thickness of 1 mm, a mylar film thickness of 250 µm, a drying temperature of 95 °C, air velocity of 0.75 m/s, and a drying time of 75 min. The dried samples were then ground with a portable electric grinder (Model: Philips, HL7756, Versuni, Kolkata, India) and were sieved through an 80-mesh sieve for the realization of RWD-processed turmeric powder. This was subjected to further characterization in a triplicate sampling mode [33].

### 2.3. Fortification with NaFeEDTA and Folic Acid

Relevant literature on turmeric fortification suggests that the fortification of finger millet flour is optimal at 6 mg of iron per 100 g of the sample [34]. Similarly, in another article, 100 g of curry powder was fortified with 20 mg of iron [22]. Also, another research group added dry folic acid to maize flour and achieved a daily intake of 100–150 µg for the acid, on a bio-accessibility basis [25]. In another work, the authors fortified salt with 1% folic acid relative to the total salt content (NaCl) [21]. Following these methods, 100 g of RW-dried turmeric powder was mixed with either 20 mg of NaFeEDTA or 20 mg of folic acid for the achievement of iron- and folic acid-fortified turmeric powder samples. In both cases, the measured amounts of folic acid and NaFeEDTA were directly mixed with RW-dried turmeric powder, and through the process of dry mixing with a spatula [34]. The experiments were carried out in triplicate.

### 2.4. Characterizations of RW-Dried Curcuma longa Powder Products

#### 2.4.1. FTIR Analysis

The FTIR spectral analysis was conducted for RW-dried turmeric powder (Model: IR Affinity, Shimadzu, Kyoto, Japan). FTIR spectrometer was operated in transmission mode was deployed for measurements conducted in the resolution range of 400–4000 cm^−1^. For the FTIR analysis, 2 g of turmeric powder sample was mixed with 300 mg of KBr. Thereafter, the thoroughly mixed sample was transformed into a pellet using a hydraulic pressure system [35].

#### 2.4.2. Thermal Analysis

The thermal transition properties of RW-dried turmeric powder, folic acid-fortified and NaFeEDTA-fortified turmeric powder samples were determined with the DSC (Model: DSC-3500 Sirius, Neitzsch, Weissenfels, Germany) and TGA systems (Model: TG 209, Libra, New York, NY, USA). Having been calibrated with indium, the instrument used an empty pan as a reference. For the measurements, a 6 mg sample was kept on to the aluminum DSC and TGA pans, and the system was subsequently sealed. From there, with a heating rate of 10 °C/min in the calorimeter, the powder samples were scanned in a temperature range of 0–600 °C and 0–1000 °C for DSC and TGA, respectively, in cases of RW-dried turmeric powder, folic acid-fortified and NaFeEDTA-fortified turmeric powder [36,37,38]. All experiments were carried out in triplicate.

#### 2.4.3. Particle Size, Poly Dispersity Index, and Zeta Potential

The average particle size, polydispersity index (PDI), and zeta potential of the RW-dried turmeric powder, folic acid fortified and NaFeEDTA-fortified turmeric powder samples were determined using a dynamic light scattering (Model: Delsa Nano C, Beckman Coulter, Brea, CA, USA) size analyzer in triplicate [36]. The procedure involved mixing 0.1 g of the sample with 20 mL of water, before subsequent sonication for 10 min. The resultant mixture was then analyzed in a cuvette at a temperature of 25 °C, with a scattering angle of 170° [37].

#### 2.4.4. Morphological Characterization

FESEM (Model: Sigma, St. Louis, MO, USA; Zeiss, Oberkochen, Germany) was deployed for the morphological characterization of the RW-dried turmeric powder, folic acid-fortified turmeric powder, and NaFeEDTA-fortified turmeric powder samples in triplicate. For this purpose, an appropriate quantity (1–3 mg) of turmeric powder was placed in the carbon tape and was coated with a thin layer (<20 nm) of gold using a sputter coater. The coated samples were investigated under 3 kV [37,38].

#### 2.4.5. XRD Analysis

The crystallographic structural analysis of RW-dried turmeric powder, folic acid-fortified turmeric powder, and NaFeEDTA-fortified turmeric powder samples were conducted using XRD (Model: Smartlab, Singapore; Rigaku Technologies, Tokyo, Japan). To do so, the instrument was operated with Cu K-alpha-1 radiation (0.154 nm), 40 kV voltage, and 30 mA current. While conducting the analysis, about 20 mg of the sample powder was loaded onto a glass plate for subsequent scanning with a Bragg angle in the range of 5 to 90° and a 0.02° per second measurement frequency [36]. All experiments were carried out in triplicate.

#### 2.4.6. In Vitro Analysis

In vitro digestion was conducted for RW-dried turmeric powder, folic acid-fortified, turmeric powder, and NaFeEDTA-fortified turmeric powder samples. A brief account of the adopted methodology has been outlined in the following sub-sections.

##### Digestion Process

The in vitro digestion process was conducted to simulate the human digestive system [39]. Thus, a two-step digestion process was followed that referred to simulated gastric and intestinal digestion processes. For the gastric digestion process, 5 g of dried powder sample was first mixed with 30 mL of 140 mM NaCl and 5 mM KCl solutions. From there, 0.5 mL of pepsin solution (11,000 U/mL) was added for subsequent stirring for 2 h in a shaking water bath kept at 37 °C. During the stirring process, 1 M HCl solution was used to maintain the system at a pH of 2. Thereafter, the sample pH was adjusted to 5 using 1 M NaHCO_3_ solution. Subsequently, the intestinal digestion procedure was carried out with the addition of 2.5 mL of pancreatin–bile solution (0.45 g of bile salts and 0.075 g of pancreatin in 37.5 mL of 0.1 M NaHCO_3_ solution) to the gastric-digested mixture. Eventually, 40 μL of 0.3 M CaCl_2_ was added, and the system pH was adjusted to 7.0 using 1 M NaOH solution. Thereafter, the system was incubated for 2 h in a shaking water bath (Model: Labtop, Mumbai, India) at 37 °C. Finally, the digested samples were cooled in ice for 10 min, and then centrifuged at 5000 rpm for 40 min at 4 °C. The final digested sample was subjected to curcumin, folic acid, and iron content analysis [40]. All experiments were carried out in triplicate.

##### Bio-Accessibility of Curcumin

For curcumin content, a calibration curve was prepared following the procedure summarized in previously published research [32,41]. Using a 10 mL volumetric flask and 95% methanol, 5 mL of digested volume was adjusted to 10 mL. Thereafter, the solution was filtered using Whatman filter paper No. 1. Subsequently, 0.4 mL of the filtrate was added to 5 mL of 95% methanol. Thereafter, the extract solution’s absorbance was measured at 425 nm in a UV/Vis spectrophotometer (Model: UV-2800, Shimadzu, Kyoto, Japan). Using the measured absorbance and a calibration chart, the curcumin content of the fresh or dried sample was determined and expressed as % curcumin content (*w*/*w*). All experiments were carried out in triplicate.

##### Bio-Accessibility of Folic Acid

For folic acid estimation, standard folic acid solution in 0.01 N NaOH was used to prepare a calibration curve, which confirmed a linear fitness plot between the folic acid concentration and the absorbance being measured. Eventually, the absorbance of the digested sample was measured at 284 nm with a UV/Vis spectrophotometer. From there, the concentration of the sample was obtained from the calibration curve [40]. All experiments were carried out in triplicate.

##### Bio-Accessibility of NaFeEDTA

After completing in vitro digestion studies, the NaFeEDTA content in the digested sample was analyzed using an atomic absorption spectrophotometer (AAS) (Model: Spectra AA 220 FS, Varian, Palo Alto, CA, USA). From there, the iron content of the sample was determined using a calibration curve, prepared with stock solutions of NaFeEDTA [42]. All experiments were carried out in triplicate.

## 3. Results and Discussion

### 3.1. FTIR

The FTIR analysis was conducted for RW-dried, folic acid-fortified, and NaFeEDTA-fortified RW-dried turmeric powder samples (Figure 1). Such studies were carried out with the intent of gaining useful insight into pertinent functional groups and discovering the influence of fortification on group shifts. The FTIR spectra of RW-dried turmeric we obtained matched those reported previously in an earlier investigation [43]. In the FTIR spectra of the dried turmeric powder, the broad band obtained at 3424 cm^−1^ was attributed to the stretching vibration of the free hydroxyl group of phenol (OH). The sharp band at 2933 cm^−1^ was attributed to the sp^2^ C‒H bond stretching. The conjugate of the carbonyl bond (C=O) with two aromatic rings was accompanied by a small shoulder at 1622 cm^−1^. The band at 600 cm^−1^ corresponded to ortho (1,2), and was substituted out of the phase of the hydroxyl and O‒CH_3_ bond at the aromatic ring. The sharp bands at 1378 and 1027 cm^−1^ were assigned to the C‒O stretch of phenyl alkyl ether. Such assignment confirmed the molecular structure of curcumin in the dried turmeric sample.

In the FTIR spectra of the folic acid-fortified RW-dried turmeric powder sample, the broad band obtained at 3421 cm^−1^ was attributed to the stretching vibration of the free hydroxyl group of phenol (OH). The band at 3427 cm^−1^ in the spectra of NaFeEDTA-fortified RW-dried turmeric powder was assigned similarly. The sharp band seen at 2926 cm^−1^ and 2934 cm^−1^ for the folic acid-fortified RW-dried turmeric powder sample and the NaFeEDTA-fortified RW-dried turmeric powder sample, respectively, was attributed to the sp^2^ C‒H bond stretching. At 1644 and 1635 cm^−1^, the conjugate of the carbonyl bond (C=O) with two aromatic rings was accompanied by a small shoulder for the folic acid-fortified RW-dried turmeric powder sample and the NaFeEDTA-fortified RW-dried turmeric powder sample, respectively.

For the folic acid-fortified RW-dried turmeric powder sample and the NaFeEDTA-fortified RW-dried turmeric powder sample, the band at 608 cm^−1^ and 622 cm^−1^, respectively, corresponded to ortho (1,2), substituted out of the phase of the hydroxyl and O‒CH_3_ bond at the aromatic ring. Also, the sharp bands seen for the folic acid-fortified RW-dried turmeric powder sample at 1386 and 1381 cm^−1^, respectively, and 1066 and 1107 cm^−1^ for the NaFeEDTA-fortified RW-dried turmeric powder sample, respectively, were attributed to the C‒O stretch of phenyl alkyl ether. Such an assignment confirms the molecular structure of curcumin in the turmeric. In the folic acid-fortified turmeric powder sample spectra, a minor shift was apparent. Similar trends also existed for the NaFeEDTA-fortified turmeric sample. These observations offer implications with regard to the stable nature of the added fortificants, and their minimal interaction with the dried turmeric powder, without any chemical reaction. Thus, with no chemical reaction, the turmeric powder did not undergo any denaturation, and this is evident in the minimal shift in the wavelength numbers seen in the functional groups. Thus, in summary, the FTIR spectral bands affirmed similar bands with minimal peak shifts.

In summary, the obtained FTIR spectra are comparable with the reported FTIR spectra for curcumin samples [43]. Also, a relevant prior study addresses FTIR analysis of curcumin in freeze-dried, hot-air-dried and sun-dried turmeric samples [44]. The peaks obtained in this prior study are in good agreement with those obtained in this work. However, marginal peak shifts did occur. This is due to the combined effects of alterations in the drying process and fortification.

### 3.2. TGA

TGA was carried out for RW-dried, folic acid-fortified RW-dried, and NaFeEDTA-fortified RW-dried turmeric powder samples (Figure 2). Figure 2a–c depicts the TGA mass-loss curve as a function of temperature. The initial temperature at which mass loss occurred was approximately 192.93 °C for the RW-dried turmeric sample, 196.42 °C for folic acid-fortified RW-dried turmeric sample, and 207.10 °C for the NaFeEDTA-fortified RW-dried turmeric sample. Thus, below these temperature values, the respective samples did not undergo degradation, and corresponding mass losses in the thermogram were due to moisture loss. Thus, the folic acid fortification did not alter the minimal temperature for degradation. This is due to the organic nature of folic acid.

Both turmeric and the folic acid-fortified turmeric sample exhibited a single stage decomposition. However, the addition of NaFeEDTA led to a multistage decomposition in the NaFeEDTA-fortified turmeric sample (mass loss at 207.10 and 567.48 °C). Regarding these, the latter is due to the presence of NaFeEDTA in the sample. In all cases, the thermal decomposition of curcumin occurred below 800 °C, and 100% mass loss was observed, confirming complete decomposition of turmeric and curcumin at elevated temperatures [45].

For the RW-dried turmeric powder sample (Figure 2a), an inflection point (broad upward peak) at about 196 °C can be observed in the TGA curve. This corresponds to the endpoint of stage I. Therefore, in this case, stage I starts at about 95 °C and ends at about 196 °C. Also, in stage I, a characteristic peak was witnessed at 144 °C. Further, stage II begins at about 200 °C and ends at about 410 °C. In this stage, a peak was witnessed at about 275 °C. Similarly, for the folic acid-fortified turmeric powder sample (Figure 2b), an inflection point (broad upward peak) at about 194 °C can be observed in the curve. This can be considered to be the endpoint of stage I. Further, stage I, starting at about 92 °C and ending at about 194 °C, is characterized with a peak at about 145 °C. Stage II begins at about 192 °C and ends at about 297 °C, and involves a peak at about 291 °C. The curves for turmeric and folic acid-fortified turmeric powder were relatively similar, and the addition of folic acid did not create any additional peaks. This is probably due to the organic nature of both compounds [46].

For the NaFeEDTA-fortified turmeric powder sample (Figure 2c), in total, seven peaks exist. These peaks have been found to be related to those obtained for turmeric and the NaFeEDTA compound in a relevant prior study [47]. The turmeric-related peak refers to the inflection point (broad upward peak) at about 196 °C in the curve. This indicated an endpoint of stage I that started at 120 °C and ended at 196 °C. The second peak was observed at 150 °C in stage I. Stage II begins at about 144 °C and ends at about 371 °C. Within this range, a fourth peak was witnessed at 292 °C. Thus, peaks numbered 1, 3, 5, and 6 were like those found for ferrous sulfate in a relevant prior study [47]. The temperatures corresponding to NaFeEDTA degradation were 107, 236, 358, and 581 °C, respectively. Thus, these peaks correspond to the degradation temperature for the NaFeEDTA compound. The seventh peak at 934 °C refers to the temperature at which the NaFeEDTA-fortified turmeric powder sample underwent degradation and complete dissolution. Thus, in summary, the analyses confirmed that, while stage I supported the decomposition of substituent groups in the curcumin, stage II involved decomposition of the curcumin due to the benzene rings of the curcumin [47,48].

### 3.3. DSC

DSC analysis was performed for the refractance window (RW)-dried, folic acid-fortified RW-dried and NaFeEDTA-fortified RW-dried turmeric powder samples (Figure 3). Figure 3a–c depict the thermograms obtained from the DSC analysis of RW-dried turmeric powder, and folic acid-fortified and NaFeEDTA-fortified RW-dried turmeric samples. For the RW-dried turmeric powder sample, an endothermic peak and exothermic peak were observed at 66.67 °C and 300.65 °C, respectively. In the case of the folic acid-fortified turmeric powder sample, the endothermic peaks were observed at 99.15 and 157.46 °C. Corresponding exothermic peaks were seen at 294.01 and 491.00 °C. It can be concluded that, at this temperature, both samples underwent crystallization. Both RW-dried turmeric and folic acid-fortified RW-dried turmeric powder samples illustrate similar trends due to their organic and amorphous nature. For the NaFeEDTA-fortified turmeric powder sample, endothermic peaks could be observed at 93.26, 151.02, 211.35, 304.29, 368.49, and 391.44 °C, and exothermic peaks could be observed at 251.12 and 475.41 °C. These peaks were attributed to the melting phase of the NaFeEDTA-fortified RW-dried turmeric powder sample. For the evaluated parameters, the NaFeEDTA-fortified RW-dried turmeric powder witnessed altered and variant findings in conjunction with the other two samples. This is due to the fortification with the crystalline natured NaFeEDTA in the RW-dried turmeric powder.

The corresponding thermograms depicted in Figure 3a–c infer that the glass transition temperatures (Tg) were 66.67, 99.15, and 93.26 °C for RW-dried turmeric powder, folic acid-fortified and NaFeEDTA-fortified RW-dried turmeric powder samples, respectively. In similar works, the authors reported a Tg value of 69.4 °C for the amorphous turmeric sample [47,48]. The T_g_ value for folic acid and NaFeEDTA is 155 and 100.20 °C, respectively [44,46]. The T_g_ value of the sample in the literature was altered due to the addition of different compounds [48]. Therefore, due to the addition of folic acid and NaFeEDTA powder, the Tg was altered for folic acid-fortified RW-dried turmeric and NaFeEDTA-fortified RW-dried turmeric powder samples.

In a relevant prior study [45], an endothermic peak was observed at 178.07 °C for turmeric extract coated with maltodextrin. This referred to a desolvation, or a loss of volatiles. In a relevant prior study, the authors obtained endothermic peaks at variant temperatures for the turmeric sample encapsulated with maltodextrin [49]. In another prior study, the researchers demonstrated endothermic peaks at similar temperatures for chitosan–turmeric sample obtained with the spray-drying process [50].

### 3.4. XRD

The XRD patterns of RW-dried turmeric powder, and folic acid-fortified and NaFeEDTA-fortified RW-dried turmeric powder samples have been illustrated, respectively, in Figure 4a–c. In Figure 4a, sharp peaks do not exist for the RW-dried turmeric powder sample. However, peaks were obtained at 17.14, 19, and 22°, with intensities of 5215, 4844, and 5079 counts per second, respectively. A peak with maximum intensity was obtained at 17.14°. Similarly, the XRD of the folic acid-fortified RW-dried turmeric powder sample (Figure 4b) indicated weak intensities. For the sample, at 11, 13.2, and 17°, the peak intensity values were 5136, 4901, and 5850 counts per second, respectively. The highest intensity was at 17°, with 5850 counts per second. For both cases, a maximum intensity was obtained at 17°. These observations agree with those found in a relevant prior study, with a maximum intensity of 5312 cycles per second at 17° [51,52].

On the other hand, the XRD pattern (Figure 4c) of NaFeEDTA-fortified RW-dried turmeric powder varied significantly in comparison to the XRD patterns of the RW-dried turmeric and folic acid-fortified RW-dried turmeric powder samples. In this case, sharp peaks at 12.64, 16.14, 21.72, and 26.44° with highest intensities of 10,984, 11,786, 7243, and 7355 counts per second were observed. The peak with maximum intensity was found at 16.14°. Similar findings have been reported for the ferrous sulfate sample at 18.3° [47].

The RW-dried turmeric and folic acid-fortified RW-dried turmeric powder samples clearly exhibited amorphous characteristics and did not create any crystalline peaks. However, the NaFeEDTA-fortified turmeric powder sample illustrated sharp peaks and, hence, indicated its crystalline nature. During RWD, rapid drying of turmeric slices took place and, thereby, the production of amorphous metastable compounds in the dried products was fostered. Thus, it can be hypothesized that for crystallization, the time was not sufficient. However, for the NaFeEDTA-fortified RW-dried turmeric case, the inorganic NaFeEDTA addition in the crystal form did contribute to the crystalline state of the product. This was not the case for the folic acid-fortified turmeric powder sample, as the folic acid has amorphous nature. The XRD of RW-dried turmeric powder and folic acid-fortified RW-dried turmeric powder samples exhibit a series of thick and intense lines, and these are indicative of their amorphous structures. On the other hand, the NaFeEDTA-fortified turmeric powder sample’s XRD patterns exhibited a series of thin and intense lines, and hence indicate a crystalline structure. These findings are also in agreement with the conclusions reached in a relevant prior study [51,52,53].

### 3.5. FESEM

The morphology of the RW-dried, folic acid-fortified RW-dried, and NaFeEDTA-fortified RW-dried turmeric powder products were studied with the FESEM instrument (Figure 5). Figure 5a depicts the structural morphology of RW-dried turmeric powder. Figure 5b,c, respectively, illustrate the structural morphology of folic acid-fortified RW-dried turmeric and NaFeEDTA-fortified RW-dried turmeric powder samples. In this regard, it should be noted that the RWD process does influence various physical properties, such as the texture, size, shape, and apparent density of the powder samples [54]. The RWD-processed samples appeared rough. This may be due to the grinding process. Further, the roughness and non-homogeneity of RW-dried turmeric, and the folic acid-fortified and NaFeEDTA-fortified RW-dried turmeric powder samples were also attributed to the partial starch gelatinization and consequent possible retrogradation during the RWD. In this regard, it should be noted that the RW-dried powder samples include starch, along with other constituents such as protein, crude fiber, and fat [52].

The porous particle surface observed in all sample morphologies was due to the melting of starch granules in the course of the disruption in the protein–starch matrix. The findings were in good agreement with those inferred in a relevant prior study [53,54]. The obtained microstructure had a similar morphology to that of turmeric powder, as reported in a recent study, and to that of a turmeric waste sample of a biorefinery [55]. From the depicted images, it can be concluded that the addition of folic acid and NaFeEDTA did not alter the morphology of the RW-dried turmeric powder. This is due to the non-degradation of the sample after fortification. To a certain extent, a smoother structure was observed in folic acid-fortified RW-dried turmeric and NaFeEDTA-fortified RW-dried turmeric powder samples. Such structures did not adhere to the turmeric powder system. The smooth structure of folic acid could not be significantly distinguished from that of the turmeric powder. This is due to the amorphous nature of both folic acid and the turmeric powder system. The NaFeEDTA powder could be distinguished from the turmeric powder in terms of morphology; the same can be said with regard to the crystalline nature of the NaFeEDTA compound.

### 3.6. Particle Size Distribution, Polydispersity Index, and Zeta Potential of Turmeric Powder Products

The average mean diameter of RW-dried turmeric, folic acid-fortified RW-dried turmeric, and NaFeEDTA-fortified RW-dried turmeric powder samples were about 1599, 1560, and 1543 nm, respectively (Table 1). This indicates that the particle size of the samples is in the micro particle size range. Furthermore, the corresponding polydispersity index values were 0.29, 0.27, and 0.25, respectively. These values affirmed excellent particle distribution and homogenous particle mean diameters. Based on this, the values also confirmed that the discussed fine powders can be explored for their significantly higher water-binding characteristics and can foster higher product quality.

It is well known that smaller particle size distribution translates into higher stability of the particles. This is due to Brownian motion [37]. All samples possessed a polydispersity index value lower than 0.40, and, thereby, confirmed the relative narrow size distribution [39]. The addition of folic acid and NaFeEDTA did not significantly alter the physical characteristics of the turmeric powder. Hence, the particle size distribution and polydispersity index of folic acid-fortified RW-dried turmeric and NaFeEDTA-fortified RW-dried turmeric powder were almost like those of the RW-dried turmeric powder. Moreover, these fine samples can be used as a nutritional constituent for therapeutic purposes. Henceforth, fine powder samples can be considered as a key ingredient in functional food [56]. The corresponding zeta potential values were −24.50, −27.50, and −25.60, respectively. The zeta potentials for all the samples were negative. High absolute values for zeta potential usually indicate a higher repulsion force from particles, and emulsion stability. However, the obtained zeta potential values were lower and negative in nature. Therefore, it was concluded that the samples precipitated after mixing occurred [39].

### 3.7. In Vitro Digestion-Based Bio-Accessibility Study of Curcumin, Folic Acid, and NaFeEDTA in Turmeric Powder Products

#### 3.7.1. Curcumin

An in vitro study of curcumin was carried out to evaluate the bio-accessibility of curcumin after digestion, and thereafter to compare it with the available curcumin content in the sample. Thus, it was found that, of the 4.80% *w*/*w* curcumin content of RW-dried turmeric, 3.10% *w*/*w* of the curcumin content was available in terms of bio-accessibility. The high retention of curcumin can be attributed to the binding of iron with curcumin and its confirmed effect of enhancement on the samples’ bio-accessibility. Previous studies also confirmed these synergistic bio- accessibility effects [57]. According to another similar work, most curcuminoids are released after incubation in the simulated intestinal fluid, a mixture of pancreatin and bile salts [58]. Bile salts could alter the interface environment and henceforth facilitate good lipase activity in the lipase present in pancreatin for better release of curcumin [59]. Such an increase in the release would possibly foster the formation of mixed micelles, which assist with curcumin solubilization [58].

#### 3.7.2. Folic Acid

The RW-dried turmeric powder may also contain folic acid as its key ingredient. The bio-accessibility of constituent folic acid in the RW-dried turmeric powder was insignificant. However, for the folic acid-fortified RW-dried turmeric powder sample, it was 9.77 mg/100 g.

#### 3.7.3. Iron Content

The constituent iron content in the RWD-processed turmeric powder sample was 41.7 mg/100 g. The bio-accessible iron content of the unfortified turmeric powder was 0.27 mg/100 g (nearly 5%). After the addition of 20 mg NaFeEDTA to the 100 g turmeric powder sample, the bio-accessible iron content was enhanced to 12.74 mg/100 g. Such an enhancement in the bio-accessibility of iron is due to the strong availability of the iron content after fortification. Previous studies have confirmed that the absorption of iron is enhanced after it has been consumed with the curcumin. This is due to the binding of iron with constituent curcuminoids in the turmeric system [57]. Also, previous studies confirmed that co-administration of iron with curcumin can reduce the side effects of iron fortification, which include side-effect-driven gastric inflammation [60].

From the bio-accessibility study, an estimate was obtained for the amount of curcumin, folic acid, and iron content which is accessible to the human body in both unfortified and fortified turmeric powder. The bio-accessibility study provides a confirmation of the actual amount of folic acid and iron present in the body after the digestion process. Thus, our investigation further consolidates the body of knowledge regarding the significance and necessity of the fortification of turmeric powder with these nutrients. The fortified turmeric powder shows better bio-accessibility of folic acid and iron.

## 4. Conclusions

The research addressed in this article provides insight into, and a brief analysis of, NaFeEDTA and folic acid fortification of RW-dried turmeric powder, and its physical, thermal, crystalline, and bio-accessibility characteristics. The FTIR spectra have similar patterns and affirm that no major shifts in the functional groups occurred due to the addition of folic acid and NaFeEDTA. In comparison to RW-dried turmeric powder and folic acid-fortified turmeric powder, the TGA of NaFeEDTA-fortified turmeric powder indicated the presence of an extra peak due to its inorganic nature. The RW-dried turmeric powder and folic acid-fortified turmeric powder showed similar trends in the DSC curves. This is due to the folic acid being organic and amorphous and the marginally crystalline nature of the NaFeEDTA-fortified turmeric powder. XRD results were the same for RW-dried turmeric powder and folic acid-fortified turmeric powder, but were different for NaFeEDTA-fortified turmeric powder. The FESEM images of the RW-dried turmeric powder, folic acid-fortified turmeric powder, and NaFeEDTA-fortified turmeric powder were similar. The particle size distribution was the same in all cases. The in vitro digestion analysis we conducted affirms that there has been a significant increase in the bio-accessibility of folic acid and NaFeEDTA content after fortification. From this work, it can be concluded that the addition of folic acid and NaFeEDTA to the RW-dried turmeric powder did not undesirably alter the physical characteristics and constitution of the native RW-dried turmeric powder. This is due to the stability of folic acid and the NaFeEDTA compounds which were added to the RW-dried turmeric powder. Turmeric powder serves as a stable base for the fortification of iron and folic acid. These are well known essential micronutrients which have been recommended for intake for children and pregnant women. The current study thus provides a safe and synergistic approach to the fortification of *Curcuma longa* powder with iron and folic acid. The iron-fortified turmeric powder can be mixed with milk or other beverages, and is recommended for application in dietary intake. Accordingly, the sample can act as a health supplement and effectively mitigate folic acid and iron deficiency. In the near future, the RW-dried turmeric powder should be subjected to fortification with alternate minerals and micronutrients such as zinc, calcium, vitamin A, etc. Also, alternate products such as ginger powder and sprout powder could be targeted as a base for fortification with iron and folic acid. Such research strategies, through the fortification route, may further strengthen humanity’s endeavors to create affordable, diverse, and varied nutritional supplementary foods.

## Figures and Tables

**Figure 1 foods-14-00658-f001:**
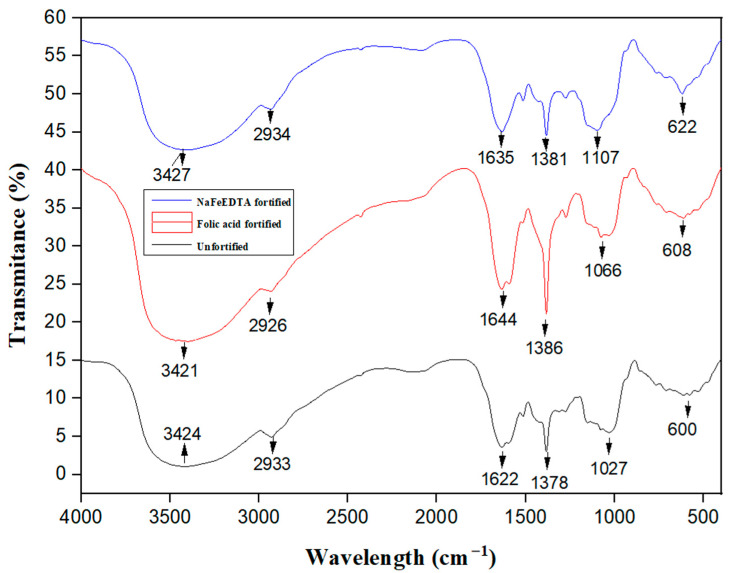
FTIR spectra of unfortified and fortified turmeric powder product samples.

**Figure 2 foods-14-00658-f002:**
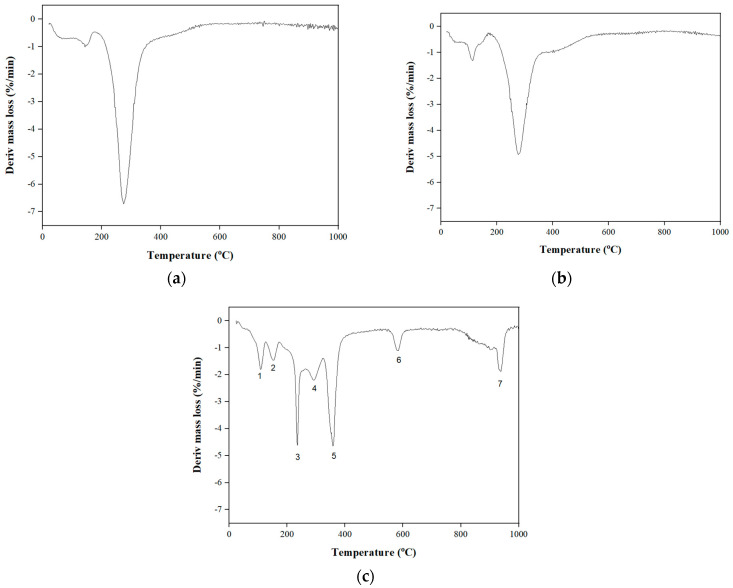
Derivative mass-loss plots for (**a**) RW-dried, (**b**) folic acid-fortified RW-dried and (**c**) NaFeEDTA-fortified RW-dried turmeric powder samples.

**Figure 3 foods-14-00658-f003:**
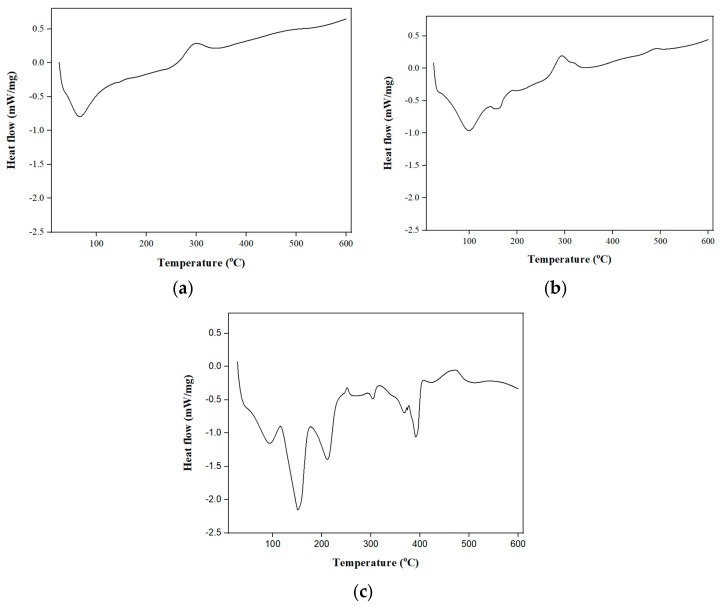
Heat flow vs. temperature plots for the (**a**) refractance window (RW)-dried, (**b**) folic acid-fortified RW-dried and (**c**) NaFeEDTA-fortified RW-dried turmeric powder samples.

**Figure 4 foods-14-00658-f004:**
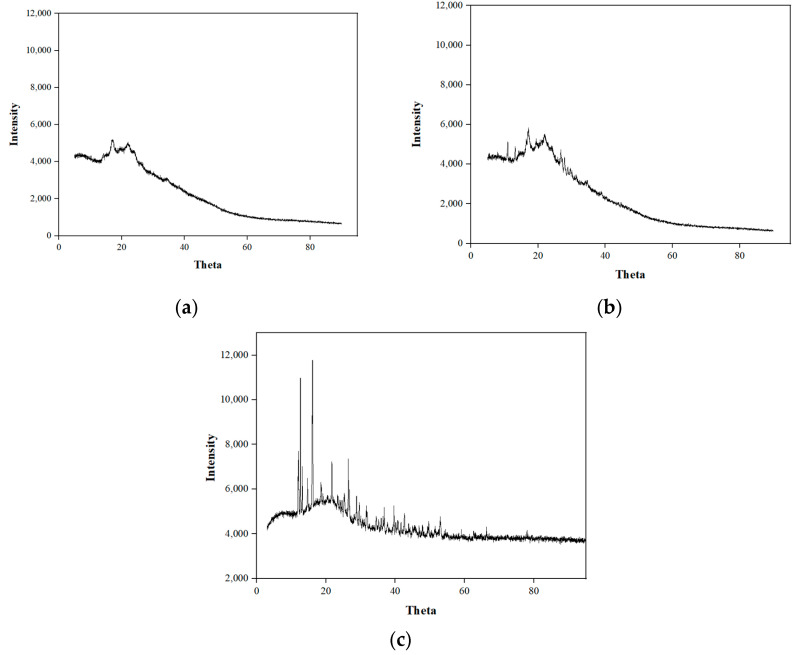
XRD spectral diagrams of the (**a**) refractance window (RW)-dried, (**b**) folic acid-fortified RW-dried and (**c**) NaFeEDTA-fortified RW-dried turmeric powder samples.

**Figure 5 foods-14-00658-f005:**
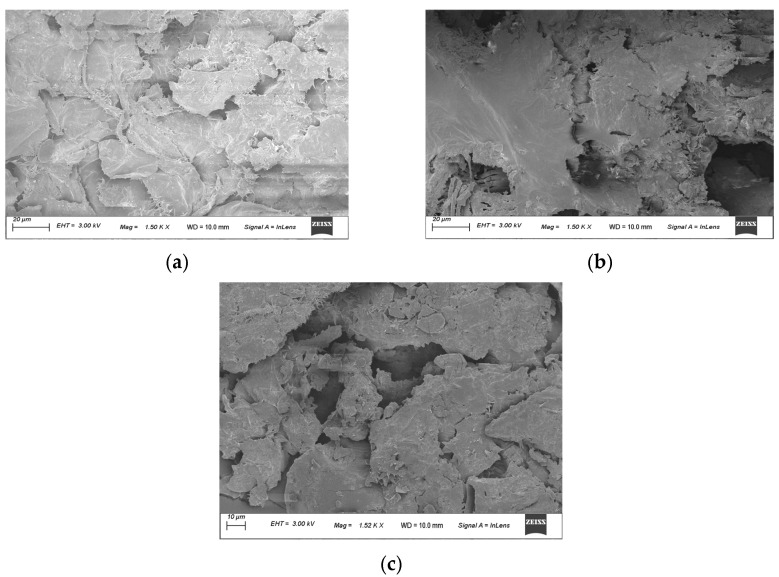
Scanning electron microscope image of (**a**) RW-dried, (**b**) folic acid-fortified RW-dried, and (**c**) NaFeEDTA-fortified RW-dried turmeric powder products.

**Table 1 foods-14-00658-t001:** Particle size, polydispersity index, and zeta potential data summary for unfortified and fortified refractance window-dried turmeric powder samples.

S. No.	Sample	Particle Size (nm)	PDI	Zeta Potential
1.	Unfortified	1599 ± 35	0.29 ± 0.05	−24.50 ± 1.2
2.	Folic acid-fortified	1560 ± 27	0.27 ± 0.03	−27.50 ± 1.8
3.	NaFeEDTA-fortified	1543 ± 18	0.25 ± 0.02	−25.60 ± 2.1

## Data Availability

The data presented in this study are available on request from the corresponding author. This is due to the ongoing work related to this project.

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
