# Peer review of "Development and Characterization of Refractance Window-Dried Curcuma longa Powder Fortified with NaFeEDTA and Folic Acid: A Study on Thermal, Morphological, and In Vitro Bio Accessibility Properties"

_foods, 2025, doi:10.3390/foods14040658_

Round 1
Reviewer 1 Report
Comments and Suggestions for Authors
The work deals with the fortification process of curcuma powder dried by refractory window and fortified to contribute to nutritional problems. The work emphasizes on morphological, thermal and bioavailability aspects.
The work is interesting and well organized, however there are opportunities for improvement.
Regarding the use of abbreviations it is important that these are conveniently defined and then used throughout the paper.
In l34 a reference is missing
l47-48 the idea can be extended.
Add a reference at the end of l56
Paragraphs on refractance window should be reversed, first talking about the technology and then about work or research in the area. Also a definition of the technology, advantages and disadvantages versus other drying methods is missing.
The wording of l91 -98 should be improved and reduced to clearly state the problem addressed in the paper.
l103 Are there any characteristics such as quality, caliber, category of the curcuma used to standardize the experiment?
l106 avoid phrases such as and other chemicals.
The drying process is incomplete, was it carried out in a continuous process, or in batches, belt speed if it was continuous.
How was the particle size measured is not specified?
l121 salt refers to NaCl?
l124- Is there any specific procedure or mixing conditions to report?
l137, 142 Standardize thermogravimetris or TGA?
Any additional information to report on the 2.4.4 conditions?
l153 what small quantity do you mean by small quantity do you need to specify?
In several parts of the methodology it uses the word make to define marks, check and correct.
When spectrophotometric measurements were performed, the equipment used is not reported.
The work does not mention replicates or statistical analysis. It is important that this is reported because its absence detracts from the validity of the results.
Important: In the presentation of results there is a theoretical preamble for each analysis, eliminate these parts. On the other hand, the results are not discussed with the necessary level of depth, they are only limited to presenting data without discussing and analyzing the results in the specific contexts.
It is important to review the scales in Figure 2, it is not always good to use auto-scaling... different scales can lead to misperception and detract from the validity of the comparisons. To give just one example, Figure 3 and 4 have 3 different scales.
It is necessary to improve the description and analysis of microscopy results.
Review the relevance of 3.7, what is the contribution as a result?
Improve the conclusion with emphasis on the findings and knowledge generated, avoid wording such as first ... second. Also include at the end of the conclusions the practical applications and possible continuation.
Author Response
Comment 1: Regarding the use of abbreviations, it is important that these are conveniently defined and then used throughout the paper.
Response: We agree with the reviewer regarding the inconsistencies of abbreviations. Accordingly, we have corrected their use for the entire manuscript thoroughly.
Comment 2: In l34 a reference is missing
Response:
We thank the reviewer for the comment. Accordingly, the reference has been incorporated. (Lines 35-36, highlighted)
In order to address these issues, food fortification has emerged as the preferred solution with respect to the traditional pharmaceutical supplements [1].
Comment 3: line 47-48 the idea can be extended.
Response:
We thank the reviewer for the comment. Accordingly following changes were incorporated. (Lines 75-85, highlighted)
Several products such as salt, wheat, corn, and rice have been fortified with iron [13, 14]. However, iron has been the most difficult micronutrient to incorporate in food products. This is due to the differences in iron absorption rates. Thereby, undesirable variations in flavor and color of the food are inevitable. Another issue could be that the inhibitors present in various food matrices could retard iron absorption. Despite such mentioned challenges, technologies have been well-established supporting the large-scale fortification of wheat and corn flour [14]. The recently developed extrusion-based premix technique for rice fortification has demonstrated significant potential [14]. However, many alternate iron-fortified food products are to be developed to affirm upon the availability of choices and varieties for a wide range of global population.
Comment 4: Add a reference at the end of line 56
Response:
The reference has been added. (Line 99, highlighted)
Comment 5: Paragraphs on refractance window should be reversed, first talking about the technology and then about work or research in the area. Also, a definition of the technology, advantages and disadvantages versus other drying methods is missing.
Response:
We thank the reviewer for the comment. Accordingly following changes were incorporated. (Lines 121-134, highlighted)
Refractance Window (RW) drying is a gentle, advanced, and energy-efficient process as it employs a thin-film drying system coupled with water recirculation for quicker and effective moisture removal from food materials. Herein, food product is spread as a thin layer on a transparent plastic conveyor belt that floats upon a heated water surface. The "refractance window" effect occurs as the infrared energy is transmitted through the belt and thereby prompts the water removal in the product that constitutes the same. This leads to efficient drying at lower product temperatures (< 80°C). In comparison to the conventional drying methods, the RWD technique offers significant advantages in terms of final product quality, including improved texture, color retention, and nutrient preservation, as well as lower energy consumption and reduced drying time [30, 32, 33]. However, RW drying has numerous challenges in comparison to the other drying methods. These are limited throughput, contingent on water availability, initial high cost, and surface drying rather deep penetration in food. Despite its constraints, the RW drying presents affordable potential for the large-scale production of high-quality food products.
Comment 6: The wording of line 91 -98 should be improved and reduced to clearly state the problem addressed in the paper.
Response:
We thank the reviewer for the comment. Accordingly following changes were incorporated. (Lines 135-150, highlighted)
So far, the parametric optimization of RW drying process has been primarily focused on carrots and onions and little attention was towards other fortified versions [31, 32]. In few studies, RW drying technique has been applied for the preservation of bioactives in Curcuma longa [28, 29]. This also includes our earlier work, wherein we optimized RW process for Curcuma longa [29]. However, extensive research on turmeric being prepared with RW drying in terms of its product fortification is confined. Thus, we hypothesized that there is a possibility to develop a new functional and bioavailable fortified turmeric product. Keeping this in mind, our hypothesis would be tested with a comprehensive assessment of the RW dried turmeric powder along with its folic acid and NaFeEDTA fortified variants in terms of thermal stability, morphological characteristics, and in-vitro bioavailability. The properties of the developed product would be evaluated with specific analytical techniques such as thermogravimetric analysis (TGA), differential scanning calorimetry (DSC), field-emission scanning electron microscopy (FESEM), X-ray diffraction (XRD), Fourier transform infrared spectroscopy (FTIR), and particle size analyzer. The developed turmeric product has prospects as a nutritional source for a large group of worldwide population.
Comment 7: l103 Are there any characteristics such as quality, caliber, category of the curcuma used to standardize the experiment?
Response:
We thank the reviewer for the comment. Accordingly following changes were incorporated. (Lines 153 - 156, highlighted)
The local turmeric variety (Curcuma longa) being cultivated in Assam, India, was sourced from the vendor at the market complex in Indian Institute of Technology Guwahati, Kamrup, Assam, India. Thereafter, it was packaged in a polyethylene pouch to prevent contamination during transportation.
Comment 8: l106 avoid phrases such as and other chemicals.
Response:
We thank the reviewer for the comment. Accordingly, the phrase has been deleted (Line 156 - 160, highlighted)
Sodium ferric ethylenediaminetetraacetate (NaFeEDTA), folic acid, potassium bromide (KBr), sodium chloride (NaCl), potassium chloride (KCl), hydrochloric acid (HCl), sodium bicarbonate (NaHCO3), calcium chloride (CaCl2), sodium hydroxide (NaOH), pepsin, pancreatin, bile salts and curcumin were obtained from Sigma Aldrich India.
Comment 9: The drying process is incomplete, was it carried out in a continuous process, or in batches, belt speed if it was continuous.
Response:
We thank the reviewer for the comment. Accordingly following changes were incorporated. (Lines 165-170 highlighted)
The RWD process was carried out in batch process and with optimized parameters from prior experiments i.e., a sample thickness of 1 mm, mylar film thickness of 250 µm, drying temperature of 95°C, air velocity of 0.75 m/s, and a drying time of 75 minutes. The dried samples were then ground with a portable electric grinder and were sieved through an 80 mesh sieve for the realization of RWD processed turmeric powder. This was subjected for further characterization in triplicate sampling mode [34].
Comment 10: How was the particle size measured is not specified?
Response:
We thank the reviewer for the comment. Accordingly following changes were incorporated. (Lines 206-209 highlighted)
The procedure involved mixing 0.1 g of the sample with 20 mL of water, and subsequent sonication for 10 minutes. The resultant mixture was then analyzed in a cuvette at a temperature of 25 °C, and with a scattering angle of 170° [38].
Comment 11: l121 salt refers to NaCl?
Response:
We thank the reviewer for the comment. Accordingly following changes were incorporated. (Line 176-177, highlighted)
In another work, the authors fortified salt with 1% folic acid relative to the total salt content (NaCl) [21].
Comment 12: l124- Is there any specific procedure or mixing conditions to report?
Response:
We thank the reviewer for the comment. Accordingly following changes were incorporated. (Lines 180 – 182, highlighted)
In both cases, the measured amount of folic acid and NaFeEDTA were directly mixed with RW dried turmeric powder and through the process of dry mixing with a spatula [22]. The experiments were carried out in triplicates.
Comment 13: l137, 142 Standardize thermogravimetries or TGA?
Response:
We thank the reviewer for the comment. TGA stands for thermogravimetric analysis. Accordingly, it is standardised in the manuscript.
Comment 14: Any additional information to report on the 2.4.4 conditions?
Response: We thank the reviewer for the comment. Accordingly following changes were incorporated. (Lines 206-209, highlighted)
The procedure involved mixing 0.1 g of the sample with 20 mL of water, and subsequent sonication for 10 minutes. The resultant mixture was then analyzed in a cuvette at a temperature of 25 °C, and with a scattering angle of 170° [38].
Comment 15: l153 what small quantity do you mean by small quantity do you need to specify?
Response: We thank the reviewer for the comment. Accordingly following changes were incorporated. (Lines 213 – 215, highlighted)
To do so, an appropriate quantity (1-3 mg) of turmeric powder was placed in the carbon tape and was coated with a thin layer (< 20 nm) of gold using a sputter coater. The coated samples were investigated under 3 kV [39-40].
Comment 16: In several parts of the methodology it uses the word make to define marks, check and correct.
Response: We thank the reviewer for pointing this out. Accordingly, it has been corrected in the entire manuscript.
Comment 17: When spectrophotometric measurements were performed, the equipment used is not reported.
Response: We thank the reviewer for pointing this. Accordingly, the corrections have been included. (Line 251, highlighted)
Comment 18: The work does not mention replicates or statistical analysis. It is important that this is reported because its absence detracts from the validity of the results.
Response: We thank the reviewer for the comment. Accordingly, we have incorporated the following changes.
Table 1. Particle size, polydispersity index and zeta potential data summary for unfortified and fortified refractance window dried turmeric powder samples.
S. No. |
Sample |
Particle size (nm) |
PDI |
Zeta potential |
1. |
Unfortified |
1599 ± 35 |
0.29 ± 0.05 |
-24.50 ± 1.2 |
2. |
Folic acid fortified |
1560 ± 27 |
0.27 ± 0.03 |
-27.50 ± 1.8 |
3. |
NaFeEDTA fortified |
1543 ± 18 |
0.25 ± 0.02 |
-25.60 ± 2.1 |
Comment 19: Important: In the presentation of results there is a theoretical preamble for each analysis, eliminate these parts. On the other hand, the results are not discussed with the necessary level of depth, they are only limited to presenting data without discussing and analyzing the results in the specific contexts.
Response: Thank you, the discussions have been carried out. (Line 270 – 532, highlighted)
3.1. FTIR Analysis
The FTIR analysis was conducted for RW dried, folic acid fortified and NaFeEDTA fortified RW dried turmeric powder samples (Figure 1). Such studies will enable useful insights into pertinent functional groups and the influence of fortification on the group shifts. The FTIR spectra of RW dried turmeric obtained matched with that reported previously in an earlier investigation [45]. In the FTIR spectra of dried turmeric powder, the broad band obtained at 3424 cm-1 was allocated to the stretching vibration of the free hydroxyl-group of phenol (OH). The sharp band at 2933 cm-1 was attributed to the sp2 C‒H bond stretching. The conjugate of carbonyl bond (C=O) with two aromatic rings was accompanied by a small shoulder at 1622 cm-1. The band at 600 cm-1corresponded to ortho (1,2), substituted out of the phase of hydroxyl and O‒CH3 bond at the aromatic ring. The sharp bands at 1378 and 1027 cm-1 were assigned to the C‒O stretch of phenyl alkyl ether. Such assignment affirmed upon the molecular structure of curcumin in the dried turmeric sample.
In the FTIR spectra of folic acid fortified RW dried turmeric powder sample, the broad band obtained at 3421 cm-1 was allocated to the stretching vibration of the free hydroxyl-group of phenol (OH). Similar assignment was for the band at 3427 cm-1 in the spectra of NaFeEDTA fortified RW dried turmeric powder. The sharp band seen at 2926 cm-1 and 2934 cm-1 for folic acid fortified RW dried turmeric powder sample and NaFeEDTA fortified RW dried turmeric powder sample respectively was attributed to the sp2 C‒H bond stretching. At 1644 and 1635 cm-1, the conjugate of carbonyl bond (C=O) with two aromatic rings was accompanied by a small shoulder for folic acid fortified RW dried turmeric powder sample and NaFeEDTA fortified RW dried turmeric powder sample respectively.
For folic acid fortified RW dried turmeric powder sample and NaFeEDTA fortified RW dried turmeric powder sample, the band at 608 cm-1 and 622 cm-1 respectively corresponded to ortho (1,2), substituted out of the phase of hydroxyl and O‒CH3 bond at the aromatic ring. Also, the sharp bands for folic acid fortified RW dried turmeric powder sample at 1386 and 1381 cm-1 respectively and 1066 and 1107 cm-1 for the NaFeEDTA fortified RW dried turmeric powder sample respectively were assigned to the C‒O stretch of phenyl alkyl ether. Such assignment confirmed upon the molecular structure of curcumin in the turmeric. In the folic acid fortified turmeric powder sample spectra, a minor shift was apparent. Similar trends also existed for the NaFeEDTA fortified turmeric sample. These observations infer upon the stable nature of the added fortificants, and their minimal interaction with the dried turmeric powder and without any chemical reaction. Thus, with no chemical reaction, the turmeric powder did not undergo any denaturation, and this has been evident in the minimal shift in the wavelength numbers of the functional groups. Thus, in summary, the FTIR spectral bands affirmed similar bands and with minimal peak shifts.
In summary, the obtained FTIR spectra have been comparable with the reported FTIR spectra for the curcumin samples [45]. Also, a relevant prior art addresses FTIR analysis of curcumin in the freeze dried, hot-air dried and sun-dried turmeric samples [46]. The peaks obtained in this prior art have been in good agreement with those obtained in this work. However, marginal peak shifts did occur. This is due to the combined effect of alterations in the drying process and fortification.
3.2. TGA Analysis
TGA was carried out for for RW dried, folic acid fortified RW dried and NaFeEDTA fortified RW dried turmeric powder samples (Figure 2). Figure 2 (a – c) depict the TGA mass loss curve as a function of temperature. The initial temperature of the mass loss was approximately 192.93°C for the RW dried turmeric sample, 196.42°C for folic acid fortified RW dried turmeric sample and 207.10 oC for the NaFeEDTA fortified RW dried turmeric sample. Thus, below these temperature values, the respective samples did not undergo degradation and corresponding mass losses in the thermogram were due to moisture loss. Thus, the folic acid fortification did not alter the minimal temperature for degradation. This is due to the organic nature of folic acid.
Both turmeric and folic acid fortified turmeric sample exhibited a single stage decomposition. However, the addition of NaFeEDTA led to a multistage decomposition in the NaFeEDTA fortified turmeric sample (mass loss at 207.10 and 567.48 °C). Among these, the latter is due to the presence of NaFeEDTA in the sample. For all cases, the thermal decomposition of curcumin occurred below 800 °C and 100 % mass loss was observed and this confirmed complete decomposition of turmeric and curcumin at elevated temperature [47].
For the RW dried turmeric powder sample (Figure 2a), an inflection point (broad upward peak) at about 196 °C can be observed in the TGA curve. This corresponds to the end-point of stage I. Therefore, for the case, stage I starts at about 95 °C and ends at about 196°C. Also, the stage I witnessed a characteristic peak at 144 °C. Further, stage II begins at about 200°C and ends at about 410 °C. This stage witnessed a peak at about 275 °C. Similarly, for the folic acid fortified turmeric powder sample (Figure 2b), an inflection point (broad upward peak) at about 194 °C can be observed in the curve. This can be considered as the end-point of stage I. Further, the stage I that starts at about 92 °C and ends at about 194°C has been characterized with a peak at about 145 °C. Stage II begins at about 192°C and ends at about 297 °C and involves a peak at about 291 °C. The curves for both turmeric and folic acid fortified turmeric powder were almost similar and the addition of folic acid did not infer upon any additional peaks. This is probably due to the organic nature of both compounds [48].
For the NaFeEDTA fortified turmeric powder sample (Figure 2c), in total, seven peaks exist. These peaks have been inferred to those obtained for turmeric and NaFeEDTA compound in a relevant prior art [54]. The turmeric related peak refers to the inflection point (broad upward peak) at about 196 °C in the curve. This indicated an end point of stage I that started at 120 °C and ended at 196°C. The second peak was observed at 150 °C in stage I. Stage II begins at about 144°C and ends at about 371 °C. The region witnessed the fourth peak at 292 °C. Thus, peaks numbered 1, 3, 5 and 6 were like those inferred for ferrous sulphate in a relevant prior art [54]. The temperatures corresponding to NaFeEDTA degradation were 107, 236, 358 and 581 °C respectively. Thus, these peaks correspond to the degradation temperature for the NaFeEDTA compound. The seventh peak at 934 °C refers to the temperature at which the NaFeEDTA fortified turmeric powder sample underwent degradation and complete dissolution. Thus, in summary, the analyses confirmed that while stage I corroborated to the decomposition of substituent groups in the curcumin, stage II involved decomposition due to benzene rings of the curcumin [54].
3.3. DSC Analysis
DSC analysis was performed for the refractance window (RW) dried, folic acid fortified RW dried and NaFeEDTA fortified RW dried turmeric powder samples (Figure 3). Figure 3 (a – c) depict the thermograms obtained from the DSC analysis of RW dried turmeric powder, folic acid fortified and NaFeEDTA fortified RW dried turmeric samples. For the RW dried turmeric powder sample, the endothermic peak and exothermic peak were observed at 66.67 °C and 300.65 °C respectively. For the case of folic acid fortified turmeric powder sample, the endothermic peaks can be observed at 99.15 and 157.46 °C. Corresponding exothermic peaks were at 294.01 and 491.00 °C. It can be concluded that at this temperature, both samples underwent crystallization. Both RW dried turmeric and folic acid fortified RW dried turmeric powder samples illustrate similar trends due to their organic and amorphous nature. For the NaFeEDTA fortified turmeric powder sample, the endothermic peaks could be observed at 93.26, 151.02, 211.35, 304.29, 368.49 and 391.44 °C and exothermic peaks at 251.12 and 475.41 °C. These peaks have been attributed to the melting phase of NaFeEDTA fortified RW dried turmeric powder sample. For the evaluated parameters, the NaFeEDTA fortified RW dried turmeric powder witnessed altered and variant findings in conjunction with the other two samples. This is due to the fortification with the crystalline natured NaFeEDTA in the RW dried turmeric powder.
The corresponding thermograms depicted in Fig 3 (a – c) infer that the glass transition temperature (Tg) was 66.67, 99.15 and 93.26 °C for RW dried turmeric powder, folic acid fortified and NaFeEDTA fortified RW dried turmeric powder samples respectively. In similar works, the authors reported a Tg value of 69.4 °C for the amorphous turmeric sample [44]. The Tg value for folic acid and NaFeEDTA is 155 and 100.20 °C respectively [47-48]. The Tg value of the sample in the literature altered due to the addition of different compounds [50]. Therefore, due to addition of folic acid and NaFeEDTA powder, the Tg altered for folic acid fortified RW dried turmeric and NaFeEDTA fortified RW dried turmeric powder samples.
In a relevant prior art [51], an endothermic peak was observed at 178.07 °C for turmeric extract coated with maltodextrin. This referred to a desolvation or loss of volatiles. In a relevant prior study, the authors obtained endothermic peaks at variant temperatures for the turmeric sample encapsulated with maltodextrin [52]. In another prior art, the researchers demonstrated endothermic peaks at similar temperatures for chitosan-turmeric sample obtained with the spray drying process [53].
3.4. XRD Analysis
The XRD patterns of RW dried turmeric powder, folic acid fortified and NaFeEDTA fortified RW dried turmeric powder samples have been illustrated respectively in Figure 4(a – c). In Figure 4 (a), sharp peaks do not exist for the RW dried turmeric powder sample. However, peaks have been obtained at 17.14, 19 and 22 º with intensities of 5215, 4844 and 5079 count per second respectively. The peak with maximum intensity has been obtained at 17.14º. Similarly, the XRD of folic acid fortified RW dried turmeric powder sample (Figure 4(b)) indicated weak intensities. For the sample, at 11, 13.2 and 17º, the peaks intensity values were 5136, 4901 and 5850 count per second respectively. Highest intensity has been at 17º and with 5850 count per count. For both cases, the maximum intensity was obtained at 17º. These observations agree with those inferred in a relevant prior art with maximum intensity of 5312 cycles per second at 17º [54].
On the other hand, the XRD pattern (Figure 4 c) of NaFeEDTA fortified RW dried turmeric powder varied significantly in comparison to the XRD patterns of RW dried turmeric and folic acid fortified RW dried turmeric powder samples. For this case, sharp peaks at 12.64, 16.14, 21.72 and 26.44 º with highest intensities of 10984, 11786, 7243, and 7355 count per second have been observed. The peak with maximum intensity has been achieved at 16.14 º. Similar findings have been reported for the ferrous sulphate sample at 18.3 º [57].
The RW dried turmeric and folic acid fortified RW dried turmeric powder samples clearly exhibited amorphous characteristics and did not confirm upon any crystalline peaks. However, the NaFeEDTA fortified turmeric powder sample illustrated sharp peaks and hence its crystalline nature. During RWD, rapid drying of turmeric slices took place and thereby fostered the production of amorphous metastable compounds in dried products. Thus, it can be hypothesized that for crystallization, the time has not been sufficient. However, for the NaFeEDTA fortified RW dried turmeric case, the inorganic NaFeEDTA addition in the crystal form did contribute to the crystalline state of the product. This was not the case for the folic acid fortified turmeric powder sample as the folic acid has amorphous nature. The XRD of RW dried turmeric powder and folic acid fortified RW dried turmeric powder samples exhibit a series of thick and intense lines, and these are indicative of their amorphous structures. On the contrary, NaFeEDTA fortified turmeric powder sample XRD patterns exhibited a series of thin and intense lines, and hence its crystalline structure. These findings are also in agreement with the inferences deduced in a relevant prior art [54 – 57].
3.5. FESEM Analysis
The morphology of the RW dried, folic acid fortified RW dried and NaFeEDTA fortified RW dried turmeric powder products were studied with the FESEM instrument (Figure 5). Figure 5 (a) depicts the structural morphology of RW dried turmeric powder. Figure 5 (b) and (c) respectively illustrate the structural morphology of folic acid fortified RW dried turmeric and NaFeEDTA fortified RW dried turmeric powder samples. In this regard, it shall be noted that the RWD process does influence various physical properties such as texture, size, shape, and apparent density of the powder samples [58]. The RWD processed samples appeared rough. This may be due to the grinding process. Further, the roughness and non-homogeneity of RW dried turmeric, folic acid fortified and NaFeEDTA fortified RW dried turmeric powder samples have been also attributed to the partial starch gelatinization and consequent possible retrogradation during the RWD. In this regard, it shall be noted that the RW dried powder samples constitute starch along with other constituents such as protein, crude fiber, and fat.
The porous particle surface being observed in all sample morphologies has been due to the melting of starch granules in due course of the disruption in the protein-starch matrix. The findings were in good agreement with those inferred in a relevant prior art [59 – 60]. The obtained microstructure was similar to the morphology of turmeric powder reported in a recent study and for the turmeric waste sample of a biorefinery [61]. From the depicted images, it can be concluded that the addition of folic acid and NaFeEDTA did not alter the morphology of the RW dried turmeric powder. This is due to non-degradation of the sample after fortification. To a certain extent, smoother structure has been observed in folic acid fortified RW dried turmeric and NaFeEDTA fortified RW dried turmeric powder samples. Such structures did not adhere with the turmeric powder system. The smooth structure of folic acid could not be significantly distinguished from that of the turmeric powder. This is due to the amorphous nature of both folic acid and turmeric powder system. While the NaFeEDTA powder could be distinguished from the turmeric powder in terms of morphology, the same has been ascertained to the crystalline nature of the NaFeEDTA compound.
3.6. Particle Size Distribution, Polydispersity Index and Zeta Potential of Turmeric Powder Products
The average mean diameter of RW dried turmeric, folic acid fortified RW dried turmeric and NaFeEDTA fortified RW dried turmeric powder samples were about 1599, 1560 and 1543 nm respectively. This conveys that the particle size of the samples has been in the micro particle size range. Furthermore, corresponding polydispersity index values were 0.29, 0.27 and 0.25 respectively. These values affirmed excellent particle distribution and homogenous particle mean diameters. Thereby, the values confirmed that the fine powders can be explored for their significantly higher water-binding characteristics and can foster higher quality of the products.
It is well known that the smaller particle size distribution translates into higher stability of the particles. This is due to Brownian motion [62]. All samples possessed a polydispersity index value lower than 0.40 and thereby confirmed upon the pertinent narrow size distribution [63]. The addition of folic acid and NaFeEDTA did not significantly alter physical characteristics of the turmeric powder. Hence, the particle size distribution and polydispersity index of folic acid fortified RW dried turmeric and NaFeEDTA fortified RW dried turmeric powder were almost like that of the RW dried turmeric powder. Moreover, these fine samples can be used as a nutritional constituent for therapeutic purposes. Henceforth, the fine powder samples can be considered as a key ingredient in functional food [64]. The corresponding zeta potential values were -24.50, -27.50 and -25.60 respectively. The zeta potential for all the samples were negative. High absolute values of zeta potential usually refer to higher repulsion force of particles and emulsion stability. However, the obtained zeta potential values have been lower and negative in nature. Thereby, the samples have been concluded to precipitate after mixing [63].
3.7. In-vitro Digestion based Bio-accessibility study of Curcumin, Folic Acid and NaFeEDTA in Turmeric Powder Products
3.7.1. Bio-accessibility of Curcumin
An in-vitro study of curcumin was carried out to evaluate the bio-accessibility of curcumin after digestion and thereby compare it with the available curcumin content in the sample. Thus, it was found that of the 4.80 % w/w curcumin content of RW dried turmeric, 3.10 % w/w curcumin content has been available for bio-accessibility. The high retention of curcumin can be attributed to the binding of iron with curcumin and its affirmation to enhance the samples’ bio-accessibility. Previous studies as well confirmed the synergistic bioavailability effects [65]. According to another similar work, most curcuminoids are released after incubation in the simulated intestinal fluid, a mixture of pancreatin and bile salts [66]. Bile salts could alter the interface environment and henceforth facilitate good activity of lipase present in pancreatin for the better release of curcumin [67]. Such an increase in the release would possibly foster the formation of mixed micelles that assist curcumin solubilization {66].
3.7.2. Bio-accessibility of Folic Acid
The RW dried turmeric powder may also constitute folic acid as its key ingredient. The bio-accessibility of constituent folic acid in the RW-dried turmeric powder was insignificant. However, for the fortified folic acid RW dried turmeric powder sample it was 9.77 mg/ 100 g.
3.7.3. Bio-accessibility of Iron Content
The constituent iron content in the RWD-processed turmeric powder sample was 41.7 mg/100 g. The bio-accessible iron content of the unfortified turmeric powder was 0.27 mg/100 g (nearly 5 %). After 20 mg NaFeEDTA addition to 100 g turmeric powder sample, the bio-accessible iron content enhanced to 12.74 mg/100 g. Such an enhancement in the bio-accessibility of iron is due to the easy availability of the iron content after fortification. Previous studies have confirmed that the absorption of iron enhances after it is being consumed along with the Curcumin. This is due to the binding of iron with constituent curcuminoids in the turmeric system [65]. Also, previous studied confirmed that co-administration of iron with the curcumin can reduce the side effects of iron fortification which includes gastric side effect driven inflammation [68].
From the bio-accessibility study, an estimate was obtained in terms of the amount of curcumin, folic acid and iron content being accessible for the human body in both unfortified and fortified turmeric powder. The bioaccessibility study provides a confirmation on the actual amount of the folic acid and iron present in the body after the digestion process. Thus, the investigations further consolidate upon the significance and necessity of the fortification of these nutrients in turmeric powder. The fortified turmeric powder shows better bio-accessibility of folic acid and iron.
Comment 20: It is important to review the scales in Figure 2, it is not always good to use auto-scaling... different scales can lead to misperception and detract from the validity of the comparisons. To give just one example, Figure 3 and 4 have 3 different scales.
Response: We thank the reviewer for the comment. Accordingly, we have corrected all the figures (figure 2,3,4) and incorporated in the manuscript.
Comment 21: It is necessary to improve the description and analysis of microscopy results.
Response:
We thank the reviewer for the comment. Accordingly, a detailed discussion has been provided. (Lines 437 – 464, highlighted)
Comment 22: Review the relevance of 3.7, what is the contribution as a result?
Response:
We thank the reviewer for the comment. Accordingly, following changes has been incorporated in the revised manuscript. (Lines 526 – 532, highlighted)
From the bio-accessibility study, an estimate was obtained in terms of the amount of curcumin, folic acid and iron content being accessible for the human body in both unfortified and fortified turmeric powder. The bioaccessibility study provides a confirmation on the actual amount of the folic acid and iron present in the body after the digestion process. Thus, the investigations further consolidate upon the significance and necessity of the fortification of these nutrients in turmeric powder. The fortified turmeric powder shows better bio-accessibility of folic acid and iron.
Comment 23: Improve the conclusion with emphasis on the findings and knowledge generated, avoid wording such as first ... second. Also include at the end of the conclusions the practical applications and possible continuation.
Response:
We thank the reviewer for the comment. Accordingly, we have incorporated the following changes in the conclusion section of the revised manuscript. (Lines 534 – 565, highlighted)
The addressed research in this article provides a brief understanding and analysis of NaFeEDTA and Folic acid fortification with RW dried turmeric powder and in terms of its physical, thermal, crystalline, and bio-accessibility characteristics. The FTIR spectra has a similar pattern and affirmed that no major shift in the functional groups occurred due to the addition of folic acid and NaFeEDTA. In comparison to RW dried turmeric powder and folic acid-fortified turmeric powder, the TGA of NaFeEDTA-fortified turmeric powder affirmed an extra peak due to its inorganic nature. The RW-dried turmeric powder and folic acid-fortified turmeric powder showed similar trends in the DSC curves. This is due to the folic acid being organic and amorphous and marginally crystalline nature of the NaFeEDTA-fortified turmeric powder. XRD results have been the same for RW-dried turmeric powder and folic acid-fortified turmeric powder but were different for NaFeEDTA-fortified turmeric powder. The FESEM images of the RW-dried turmeric powder, folic acid-fortified turmeric powder, and NaFeEDTA-fortified turmeric powder were similar. The particle size distribution was the same for all cases. In-vitro digestion analysis conducted affirms that there has been a significant increase in the bioavailability of folic acid and NaFeEDTA content after fortification. From this work, it could be concluded that the addition of folic acid and NaFeEDTA to the RW-dried turmeric powder did not undesirably alter the physical characteristics and constitution of the native RW-dried turmeric powder. This is due to the stability of folic acid and NaFeEDTA compounds being added to the RW-dried turmeric powder. Turmeric powder serves as stable base for the fortification of iron and folic acid. These are well known essential micronutrients being recommended for intake by children and pregnant women. The current study thus provides a safe and synergistic approach to the fortification of iron and folic acid in Curcuma longa powder. The iron fortified turmeric powder can be mixed with milk or other beverages and is recommended for the dietary intake application. Accordingly, the sample can act as a health supplement and effectively mitigate folic acid and iron deficiency. In the near future, the RW-dried turmeric powder can be subjected to fortification with alternate minerals and micronutrients such as zinc, calcium, vitamin A etc. Also, alternate products such as ginger powder and sprouts powder can be targeted as a base for the iron and folic acid fortification. Such research strategies through the fortification route will further strengthen human endeavour to create affordable, diverse and variegated nutritional supplementary foods.

Reviewer 2 Report
Comments and Suggestions for Authors
The manuscript deals with the “Development and Characterization of Refractance Window Dried Curcuma longa Powder Fortified with NaFeEDTA and Folic Acid: A Study on Thermal, Morphological, and In-Vitro Bioavailability Properties.” This study aims to fill that gap by characterizing Refractance Window dried turmeric powder, as well as its folic acid and NaFeEDTA fortified variants. This review aims the integration of NaFeEDTA and folic acid into RWD Curcuma longa powder necessitates a comprehensive evaluation of the product's thermal stability, morphological characteristics, and in-vitro bioavailability. Understanding these properties is essential to ensure that the fortification process does not compromise the quality and efficacy of the final product. Moreover, assessing the bioavailability of the added micronutrients will determine the potential health benefits of the fortified Curcuma longa powder. The topic is interesting and catches the audience’s attention, but it needs to improve according to the suggested comments.
Abstract:
Lines 25 to 28: The fortified turmeric powder exhibited enhanced crystalline properties with sharp and high intensity peaks for NaFeEDTA fortified turmeric powder. In vitro digestion studies affirmed the bioavailability of the novel fortified turmeric powder with 9.77 mg/100g, 12.74 mg/100 g for folic acid and NaFeEDTA respectively. Thus, the findings confirmed non-significant influence of fortification on the desired properties of the folic acid and NaFeEDTA fortified RW dried turmeric powder product. Reconcile these sentences.
Introduction:
Lines 43-45: Folate has gained significant global attention due to its critical role in preventing early embryonic brain development disorders, such as neural tube defects. Please provide global statistics if possible.
The lacking part of the introduction is a clear statement of the research gap or need for the current study. While the introduction mentions several studies but does not explicitly state the specific gap or problem the current study aims to address. In addition, there is no direct link between the previous studies and the objectives of the current study.
Lines 349 to 356: The efficacy of newly developed foods in terms of the bio-accessibility and bioavailability parameters is very much dependent on the availability of digestion models. These models accurately simulate the complex physicochemical and physiological events that occur in the human gastrointestinal tract. There is a real need to use in-vitro models that closely mimic the physiological processes that occur during human digestion. Such models do consider several factors such as the occurrence and concentration of digestive enzymes, pH values in gastric and intestinal phases, digestion time and salt concentrations as primary. Rewrite logically.
Conclusion:
The conclusion must be curtailed.
Comments on the Quality of English Language
Minor English editing is needed.
Author Response
- Comments on Abstract:
Lines 25 to 28: The fortified turmeric powder exhibited enhanced crystalline properties with sharp and high intensity peaks for NaFeEDTA fortified turmeric powder. In vitro digestion studies affirmed the bioavailability of the novel fortified turmeric powder with 9.77 mg/100g, 12.74 mg/100 g for folic acid and NaFeEDTA respectively. Thus, the findings confirmed non-significant influence of fortification on the desired properties of the folic acid and NaFeEDTA fortified RW dried turmeric powder product. Reconcile these sentences.
Response:
We thank the reviewer for the comment. Accordingly, we have incorporated the following changes in the abstract. (Lines 27 – 29, highlighted)
Thus, the findings confirmed that there was no significant influence of fortification on the characteristics of folic acid and NaFeEDTA fortified RW dried turmeric powder product.
- Comments on Introduction:
Lines 43-45: Folate has gained significant global attention due to its critical role in preventing early embryonic brain development disorders, such as neural tube defects. Please provide global statistics if possible.
The lacking part of the introduction is a clear statement of the research gap or need for the current study. While the introduction mentions several studies but does not explicitly state the specific gap or problem the current study aims to address. In addition, there is no direct link between the previous studies and the objectives of the current study.
Response:
We thank the reviewer for the comment. Accordingly, we have incorporated the following changes in the introduction. (Lines 46 – 57, highlighted)
NTDs, arising from the incomplete closure of the neural tube within the first 4 weeks of conception, are a major cause of serious congenital disorders, affecting 0.2 to 10 per 1000 pregnancies worldwide. While Spina bifida, anencephaly, and encephalocele are the most common types of NTDs, iniencephaly and craniorachischisis are rare [7]. Maternal folate deficiency during early pregnancy is a preventable risk factor. The World Health Organization (WHO) recommends that women of reproductive age maintain red blood cell folate concentrations above 400 ng/mL to minimize NTD risk [8]. Preventative measures include folic acid supplementation (0.4 mg daily before conception until the first trimester), consumption of folate-rich foods (leafy greens, asparagus, beets, broccoli, and artichokes), and fortified foods. For NTDs prevetion, fortification of staple foods such as wheat flour, maize flour, and rice with folic acid has been implemented in nearly 60 countries as a safe and cost-effective public health strategy [9].
Lines 349 to 356: The efficacy of newly developed foods in terms of the bio-accessibility and bioavailability parameters is very much dependent on the availability of digestion models. These models accurately simulate the complex physicochemical and physiological events that occur in the human gastrointestinal tract. There is a real need to use in-vitro models that closely mimic the physiological processes that occur during human digestion. Such models do consider several factors such as the occurrence and concentration of digestive enzymes, pH values in gastric and intestinal phases, digestion time and salt concentrations as primary. Rewrite logically.
Response:
We thank the reviewer for the comment. However, this paragraph has been removed in the revised manuscript as per the “comment 1 - reviewer 1” which stated to remove all preamble paragraphs that were previously included in the results and discussions sections.
- Conclusion:
The conclusion must be curtailed.
Response:
We thank the reviewer for the comment. Accordingly, we have incorporated the following changes in the conclusion section of the revised manuscript. (Lines 534 – 566, highlighted)
The addressed research in this article provides a brief understanding and analysis of NaFeEDTA and Folic acid fortification with RW dried turmeric powder and in terms of its physical, thermal, crystalline, and bio-accessibility characteristics. The FTIR spectra has a similar pattern and affirmed that no major shift in the functional groups occurred due to the addition of folic acid and NaFeEDTA. In comparison to RW dried turmeric powder and folic acid-fortified turmeric powder, the TGA of NaFeEDTA-fortified turmeric powder affirmed an extra peak due to its inorganic nature. The RW-dried turmeric powder and folic acid-fortified turmeric powder showed similar trends in the DSC curves. This is due to the folic acid being organic and amorphous and marginally crystalline nature of the NaFeEDTA-fortified turmeric powder. XRD results have been the same for RW-dried turmeric powder and folic acid-fortified turmeric powder but were different for NaFeEDTA-fortified turmeric powder. The FESEM images of the RW-dried turmeric powder, folic acid-fortified turmeric powder, and NaFeEDTA-fortified turmeric powder were similar. The particle size distribution was the same for all cases. In-vitro digestion analysis conducted affirms that there has been a significant increase in the bioavailability of folic acid and NaFeEDTA content after fortification. From this work, it could be concluded that the addition of folic acid and NaFeEDTA to the RW-dried turmeric powder did not undesirably alter the physical characteristics and constitution of the native RW-dried turmeric powder. This is due to the stability of folic acid and NaFeEDTA compounds being added to the RW-dried turmeric powder. Turmeric powder serves as stable base for the fortification of iron and folic acid. These are well known essential micronutrients being recommended for intake by children and pregnant women. The current study thus provides a safe and synergistic approach to the fortification of iron and folic acid in Curcuma longa powder. The iron fortified turmeric powder can be mixed with milk or other beverages and is recommended for the dietary intake application. Accordingly, the sample can act as a health supplement and effectively mitigate folic acid and iron deficiency. In the near future, the RW-dried turmeric powder can be subjected to fortification with alternate minerals and micronutrients such as zinc, calcium, vitamin A etc. Also, alternate products such as ginger powder and sprouts powder can be targeted as a base for the iron and folic acid fortification. Such research strategies through the fortification route will further strengthen human endeavour to create affordable, diverse and variegated nutritional supplementary foods.
- Comments on the Quality of English Language:
Minor English editing is needed.
Response:
We thank the reviewer for the comment. Accordingly, we have reviewed the whole manuscript for minor English corrections and has been incorporated in the revised manuscript.

Round 2
Reviewer 1 Report
Comments and Suggestions for Authors
The authors made most of the changes and the document improved considerably. I am satisfied with the current status of the paper for publication in the journal.